# Suicide adverse events associated with zopiclone and eszopiclone: A pharmacovigilance analysis based on FAERS, JADER and CVARD

**Yinpeng Xu**[1], **Fang Li**[1], **Mei Zhang**[2], **Li Huang**[2*]

1 Department of Pharmacy, Ninth People's Hospital of Zhengzhou, Zhengzhou, Henan, China,
2 Department of Pharmacy, Women and Infants Hospital of Zhengzhou, Zhengzhou, Henan, China

* sunchenzhang@hhu.edu.cn

## Abstract

### Background

Suicide adverse events are a serious concern with zopiclone and eszopiclone. This study assessed the risk of such events by mining data from the FAERS, JADER, and CVARD databases.

### Methods

The analysis included reports from Q1 2004 to Q4 2024 for FAERS, April 2004 to December 2024 for JADER, and January 1991 to December 2024 for CVARD. The risk was analyzed using the reporting odds ratio (ROR), proportional reporting ratio (PRR), and Information Component (IC).

### Results

A total of 361 reports of suicide adverse events linked to zopiclone and eszopiclone were included (FAERS: 120; JADER: 44; CVARD: 197). These events encompassed completed suicide, suicide attempt, suicidal ideation, suicidal behavior, self-injurious ideation, and intentional self-injury. Suicide attempt was consistently detected as a positive signal by all three methods across all databases and classified as a medium clinical priority.Multivariable logistic regression analysis from FAERS data indicated that males had an independently increased risk of suicide associated with these drugs [OR (95% CI): 14.70 (2.64–274.75), p = 0.012]. Most suicide-related events (60.0%) occurred within 0–180 days after taking medication. based on FAERS data. Survival analysis showed no significant gender difference in time to suicide onset (p = 0.15). The hazard rate remained constant over time (Weibull shape β = 0.80). Death was the outcome in 50.0% of cases following suicide onset. The average reported doses for zopiclone (262.5 mg) and eszopiclone (25.6 mg) significantly exceeded maximum recommended dosages.

**Data availability statement:** Raw datasets obtained from public sources were analyzed in this study.The raw data analysis from the three databases has a cutoff date of October 31, 2024. FAERS data can be found at: https://fis.fda.gov/extensions/FPD-QDE-FAERS/FPD-QDE-FAERS.html; CVARD data can be found at: https://www.canada.ca/en/health-canada/services/drugs-health-products/medeffect-canada/adverse-reaction-database/canada-vigilance-online-database-data-extract.html; JADER data can be found at: https://www.info.pmda.go.jp/fukusayoudb/CsvDownload.jsp. And all relevant data extracted from the raw data are included in the manuscript and its supplementary information files.

**Funding:** Funding for this research was provided by "Scientific Research Project of Women and Infants Hospital of Zhengzhou, China(No. YNKY202409)", "China Medical Education Association "Gathering Talents" scientific research project (No.CMEAPC2024030)", and "Zhengzhou City Medical and Health Science and Technology Innovation Guidance Program Project (No. 2025YLZDJH158)". Li Huang , the host of the first two projects, contributed to the study in the following roles: Data curation, Investigation, Methodology, Project administration, Supervision, Validation, and Writing – review and editing. Yinpeng Xu, the host of the last project, contributed to the study in the following roles: Investigation, Writing – original draft, and Writing – review & editing.

**Competing interests:** The authors declare that the research was conducted in the absence of any commercial or financial relationships that could be construed as a potential conflict of interest.

## Conclusion

These findings align with published reports and clinical observations, highlighting the need for further clinical trials to investigate suicide associated with zopiclone and eszopiclone.

---

## Introduction

Zopiclone and eszopiclone are widely used worldwide for the treatment of insomnia. Both drugs are nonbenzodiazepines that have the same pharmacological effects as benzodiazepines but with enhanced selectivity for the α1 and α3 subunits of the γ-aminobutyric acid type A (GABAA) receptor, resulting in better hypnotic and weaker muscarinic effects [1]. Eszopiclone is the S-isomer of zopiclone and exerts its effects through heterodimeric modulation of GABAA receptor activity [2]. The affinity of eszopiclone for the central GABAA receptor is 50 times greater than that of zopiclone, with an effective therapeutic dose that is half as potent and a shorter time to reach peak plasma concentration [3,4].

Most of the current research on zopiclone and eszopiclone has focused primarily on assessing their efficacy, with relatively limited attention given to investigating drug safety, particularly in relation to suicide-related adverse events. Suicide is a high-risk behaviour in which individuals take various means to end their own lives deliberately or voluntarily driven by prolonged and complex psychological activities [5]. Suicide-related events include completed suicide, suicide attempt, suicidal ideation, suicidal behaviour, self-injurious ideation, and intentional self-injury. The FDA has expressed concern regarding the heightened risk of suicide associated with this particular class of drugs [6]. For example, a one-year observational study in Oslo in 2010 reported that 2 patients committed suicide after taking zopiclone [7]. In 2019, pharmacovigilance in New Zealand reported 1 patient with suicidal ideation triggered by treatment with zopiclone [8]. In 2007, the Texas Poison Control Center in the United States reported 107 (40.2%) suspected suicide attempt among 266 patients taking eszopiclone during 2005–2006 year [9]. In 2016, Pennington JG reported a patient who experienced parasomnias and subsequently attempted suicide after taking eszopiclone for only 2 days [10].

Over the past few years, researchers around the world have assessed the safety of drugs for clinical application by mining data from large databases that record spontaneous adverse events (such as the FAERS, JADER and CVARD) [11–13].These databases are publicly accessible resource that are based on voluntary adverse event reporting by health workers, consumers, pharmaceutical manufacturers and other stakeholders. Theirs primary goal are to facilitate post-marketing safety surveillance of drugs to reduce all aspects of the losses associated with adverse drug events [14,15]. This study mined and analysed the reports of suicide-related adverse events caused by zopiclone and eszopiclone in recent years to provide valuable insights into the safe use of these two drugs.

## Materials and methods

### Source and handling of data

Suicide-related adverse event data related to zopiclone and eszopiclone in this study were reconstructed based on data obtained from the FAERS for the period Q1 2004–Q4 2024,the JADER covering April 2004 to December 2024, and the CVARD from January 1991 to December 2024.

### Data processing

Adverse events related to suicide were classified and described according to the Medical Dictionary for Regulatory Activities (MedDRA) version 27.1, utilizing the System Organ Class (SOC), High Level Group Term (HLGT), High Level Term (HLT), and Preferred Term (PT) levels. The PT codes 10042464 for suicide attempt, 10010144 for completed suicide, 10051154 for self-injurious ideation, 10071456 for suicidal ideation, 10065604 for suicidal behaviour, and 10022524 for intentional self-injury were identified through MedDRA 27.1.Before conducting pharmacovigilance analysis, generic drug names were standardized. Duplication among adverse event reports was addressed by examining criteria such as report ID, sex, and country.

### Data analysis

**Pharmacovigilance disproportionality analysis.** The three major specific indices were the ROR, PRR and IC. A stronger risk of adverse event reporting was indicated by larger values of the ROR, PRR and IC, indicating a more robust statistical relationship between the target drug and target adverse event [16–19].

These algorithms were used to derive decision rules for the purpose of determining the reporting risk and/or to compute scores for quantifying drug–adverse event associations based on a two-by-two frequency table that captures drug presence or absence and event occurrence in case reports [20–22]. S1 and S2 Tables show the formulas for the ROR, PRR, and IC calculations and two-by-two contingency table.

**Prioritization of relevant disproportionality signals.** Adverse events associated with zopiclone and eszopiclone that met the threshold for signal detection in at least one disproportionality analysis were prioritized using a semiquantitative scoring system based on the criteria detailed in Table 1. Clinical relevance was evaluated based on inclusion in the European Medicines Agency Important Medical Event (IME) and Designated Medical Event (DME) lists. DME are defined as rare but potentially drug-induced serious events. The reporting rate was defined as the proportion of reports for the target adverse event compared to all other adverse events. This was categorized using traditional frequencies: very common (≥10%), common (1%−10%), and uncommon (≤1%). The reported mortality rate was defined as the proportion of reports documenting a fatal outcome among all reports for the target adverse event. Adverse events were assigned an overall priority score based on a total score: 0–2 indicated low priority, 3–5 moderate priority, and 6–8 high priority [23].

**Logistic regression analysis.** We performed univariable and multivariable logistic regression analyses. The independent variables included age, gender, weight. Age was categorized into three groups: < 18 years old,18–65years

**Table 1. Disproportionality analysis yielded criteria and relevant scores for prioritizing adverse events.**

| Criterium | 2 points | 1 point | 0 point |
|---|---|---|---|
| Reporting rate(cases/non-cases) | > 10% | 1–10% | 0–1% |
| Signal stability(consistency across disproportionality analyses) | 3 of 3 | 2 of 3 | 1 of 3 |
| Reported case fatality rate (proportion of reports with death asoutcome) | > 50% | 25–50% | < 25% |
| Clinical relevance (serious likely drug-attributable AEs) | DME | IME | None |

old,and ≥65 years old. Gender was categorized into two groups: female and male. A criterion for statistical significance was set at P < 0.05.

**Time to onset (TTO) analysis.** As defined in this study, the time to onset (TTO) for suicide following zopiclone and eszopiclone was the interval between the EVENT_DT date recorded in the DEMO file and the START_DT date recorded in the THER file.Cases lacking complete date information were omitted. Additionally, outliers and anomalies were excluded to improve the accuracy and reliability of the TTO analysis. The Weibull distribution test was employed to effectively identify and estimate fluctuations in suicide risk incidence over time [11]. This test, characterized by its scale (α) and shape (β) parameters, allows for the identification and prediction of changes in adverse event (AE) risk incidence over time, with the shape parameter β being the primary focus here. If β is less than 1, and its 95% confidence interval (CI) is also below 1, the risk of adverse effects is considered to diminish over time, reflecting an early failure-type curve. In contrast, if β is close to 1 and its 95% CI encompasses 1, the risk is considered constant over time, corresponding to a random failure-type curve. Finally, if β exceeds 1 and its 95% CI does not include 1, the hazard is interpreted as rising over time, indicating a wear-out failure-type curve.

## Results

### Results of pharmacovigilance analysis

**Data extraction results.** A total of 21,838,627 adverse event reports were received from the FAERS database covering the period from Q1 2004 to Q4 2024. Following data mining and deduplication, 120 reports related to suicide associated with zopiclone and eszopiclone were ultimately identified (detailed in Fig 1).

**Descriptive overview of cases.** Based on the demographic information of suicide reports associated with zopiclone and eszopiclone, the results showed a larger number of female patients (56 cases, 46.7%). The most frequent age group was 18–64 years (64 cases, 53.3%). A considerable proportion of reports indicated Death as an outcome (60 cases, 50.0%). The top five reporting countries were the United States (70 cases, 58.3%), followed by Japan (32 cases, 26.7%), Germany (6 cases, 5.0%), Austria (3 cases, 2.5%), and Canada (3 cases, 2.5%). Detailed demographic statistics are presented in Table 2. Fig 2 illustrates the general trend of the number of reports submitted annually, which showed a general increase over the years.

**Disproportionality analysis.** In the FAERS database, 120 suicide associated with zopiclone and eszopiclone reports were identified. All three detection methods yielded positive signals, as presented in Table 3. Fig 3 displays a volcano plot for the identified suicide adverse events associated with zopiclone and eszopiclone. The y-axis is scaled as−log 10(P−value), representing the p-value obtained from Fisher's exact test and Bonferroni correction. A value of P = 0.05 corresponds to−log 10(P−value)≈1.3; thus larger values on the y-axis indicate a more significant difference. The x-axis is scaled as log(ROR). The color of the points represents the number of case reports, with redder colors indicating a higher number of reports. Therefore, suicide adverse events located above the dashed line in the figure exhibit significant signal strength and difference.

**Clinical prioritization of relevant disproportionality signals.** Table 4 displays the clinical priority analysis of zopiclone and eszopiclone-associated suicide in the FAERS database, based on four aspects: clinical relevance, reporting rate, fatality rate, and signal stability. The analysis classifys suicide attempt、suicidal ideation and intentional self-injury as a moderate priority adverse event, classifys completed suicide as a high priority adverse event. This study confirms the irreplaceable and supplementary role of post-marketing surveillance in detecting rare suicide adverse events associated with zopiclone and eszopiclone, thereby supporting proactive clinical monitoring by physicians and promoting safer, more rational drug use.

**Logistic regression analysis.** Findings from the univariate logistic regression analysis indicated that age was not an influencing factor for suicide associated with zopiclone and eszopiclone, with no statistically significant difference (P > 0.05). Gender was significantly associated with suicide occurrence, with a statistically significant difference at the P < 0.05 level. To further investigate the associated risk factors for suicide, multivariate logistic regression analysis was

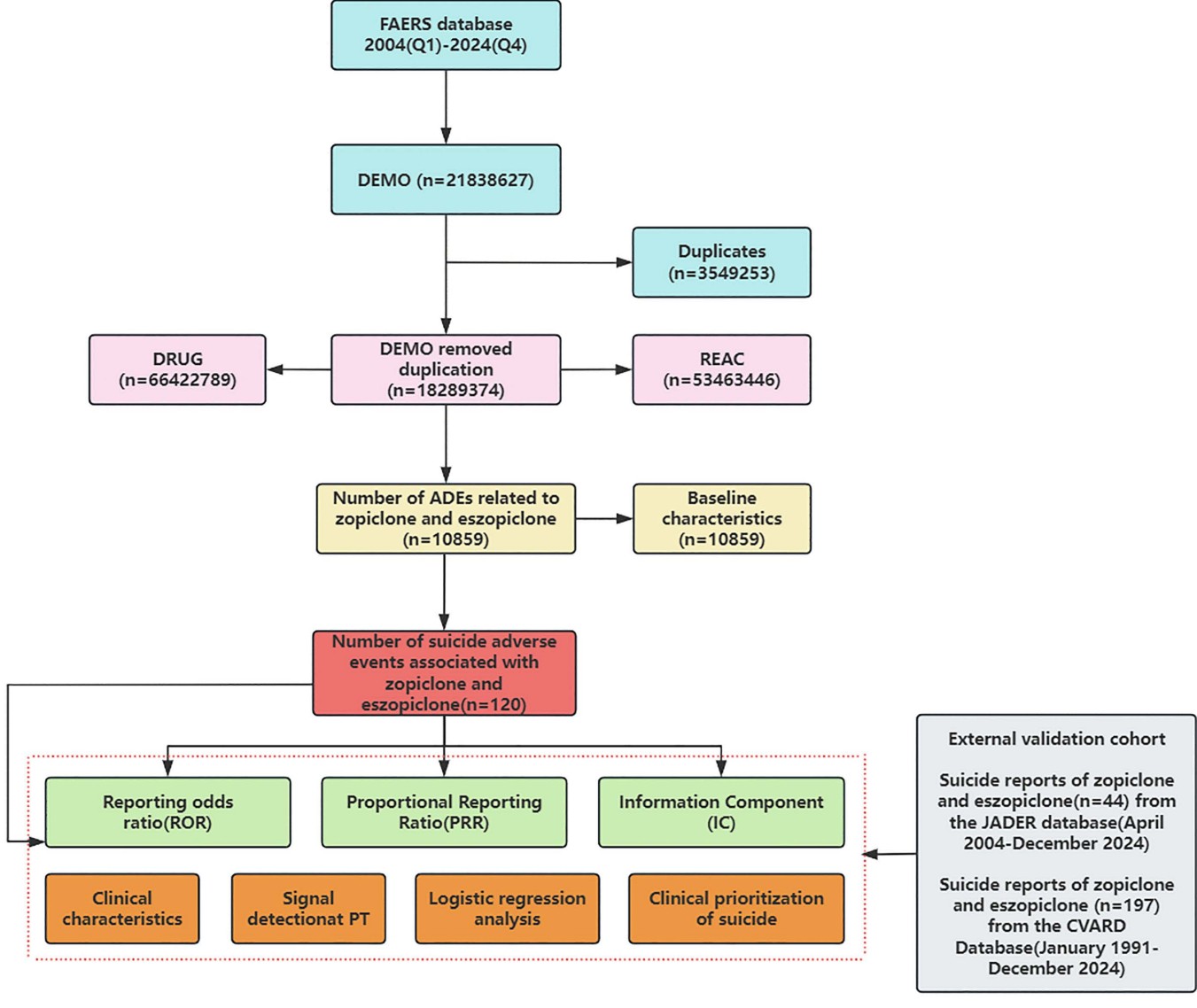

**Fig 1. The complete flowchart for the study.**

conducted. This analysis revealed that the males had a significantly higher risk of death than females, which revealed an adjusted OR of 19.17(95% CI = 3.02380.44, P = 0.009). Table 5 presents detailed results.

**TTO analysis of suicide associated with zopiclone and eszopiclone.** A total of 10 reports with available time-to-onset information for suicide following zopiclone and eszopiclone administration were identified in the FAERS database. Analysis revealed that the majority of suicide cases (6 reports, 60.0%) occurred within 0–180 days after taking medication..

Survival analysis indicated no statistically significant difference in time-to-onset of suicide between male and female recipients (p = 0.15 > 0.05).

Weibull distribution fitting was applied to the reports with available onset data. The median (Min,Max) time-to-onset was 155.5 (1,487) days.The shape parameter (β) for the onset time of suicide was estimated to be 0.80, with a 95% CI of 0.39–1.21, suggesting the risk was considered constant over time, corresponding to a random failure-type curve.

**Table 2. Characteristics of suicide reports associated with zopiclone and eszopiclone from FEARS.**

| Clinical characteristics | Zopiclone(N = 11) | Eszopiclone(N = 109) | Total(N = 120) |
|---|---|---|---|
| **Gender** | | | |
| Male | 3 | 43 | 46 |
| Female | 6 | 50 | 56 |
| Missing | 2 | 16 | 18 |
| **Age(year)** | | | |
| Median [Min, Max] | 34.67[18 - 44] | 53.45[18 - 91] | 52.79[18 - 91] |
| <18 | 0 | 0 | 0 |
| 18-64 | 3 | 61 | 64 |
| ≥65 | 0 | 21 | 21 |
| Missing | 8 | 27 | 35 |
| **Outcome** | | | |
| Death | 7 | 53 | 60 |
| Hospitalization-initial or prolonged | 3 | 29 | 32 |
| Life-threatening | 1 | 8 | 9 |
| Unknown | 0 | 1 | 1 |
| Other outcome | 0 | 18 | 18 |
| **Reported Countries** | | | |
| Germany | 6 | – | 6 |
| Sweden | 2 | – | 2 |
| France | 2 | – | 2 |
| Netherlands | 1 | – | 1 |
| United States | – | 70 | 70 |
| Japan | – | 32 | 32 |
| Austria | – | 3 | 3 |
| Canada | – | 3 | 3 |
| Great Britain | – | 1 | 1 |

**Dosage analysis of suicide associated with zopiclone and eszopiclone.** Dosage data were collected from patients who experienced suicide-related adverse events while using zopiclone and eszopiclone. In total, 5 valid zopiclone cases were extracted, with a median dose of 262.5 mg,ranging from 75 mg to 300 mg. 30 valid eszopiclone cases were extracted, with a median dose of 25.6 mg,ranging from 1 mg to 160 mg.

**External validation in JADER/ CVARD database.** To validate the findings from the FAERS, data from the JADER and CVARD databases were utilized. A total of 44 reports of suicide associated with zopiclone and eszopiclone was collected from the JADER from April 2004 to December 2024, and 197 reports were collected from the CVARD from January 1991 to December 2024. In both databases, all three detection methods indicated six positive signals for suicide associated with zopiclone and eszopiclone (e.g.,:completed suicide, suicide attempt, suicidal ideation, suicidal behaviour, self-injurious ideation, and intentional self-injury), as shown in Table 6. Furthermore, the clinical priority of suicide associated with zopiclone and eszopiclone in both databases was consistently classified as moderate priority,excepted self-injurious ideation was classified as low priority(see Table 7). Demographic statistics regarding the age and gender of reported patients are presented in Fig 4A-4B. Fig 4C-4D displays volcano plots for suicide associated with zopiclone and eszopiclone in the JADER and CVARD databases. Suicide attempt was detected as a positive signal by all three signal detection methods, for two drugs, across all three databases. Fig 5 shows the relationship between suicide-related adverse events associated with zopiclone and eszopiclone and the hierarchy in the MedDRA.Survival analysis indicated

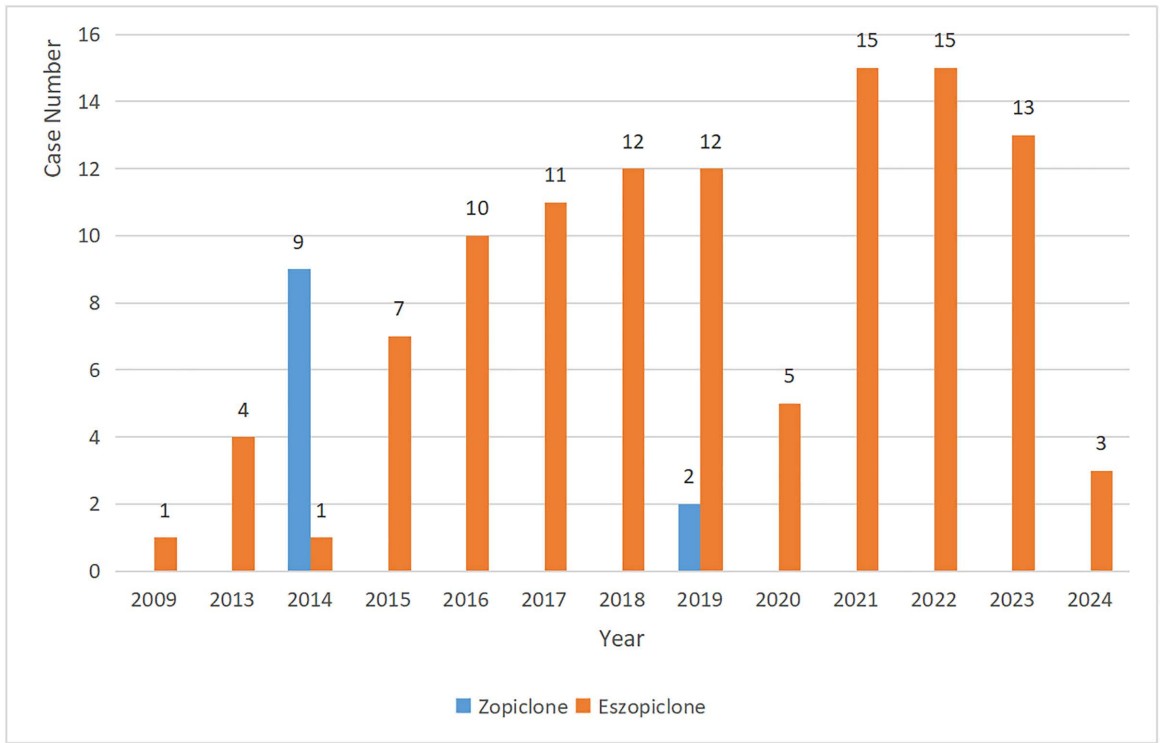

**Fig 2. Yearly frequencies of suicide reports associated with zopiclone and eszopiclone from FEARS.**

**Table 3. Detection of suicide signals associated with zopiclone and eszopiclone from FEARS.**

| Drug | PT | N | ROR (95% CI) | PRR (χ2) | IC (IC$_{025}$) |
|---|---|---|---|---|---|
| Zopiclone | suicide attempt | 4 | 12.72 (4.74 - 34.11)* | 12.57 (42.63)* | 3.65 (2.35)* |
| | completed suicide | 7 | 15.99 (7.56 - 33.83)* | 15.65 (96.16)* | 3.97 (2.94)* |
| Eszopiclone | suicide attempt | 39 | 10.24 (7.47 - 14.04)* | 10.15 (321.69)* | 3.34 (2.88)* |
| | completed suicide | 53 | 9.94 (7.58 - 13.04)* | 9.81 (419.89)* | 3.29 (2.9)* |
| | suicidal ideation | 13 | 2.24 (1.3 - 3.86)* | 2.24 (8.89)* | 1.16 (0.39)* |
| | intentional self-injury | 4 | 3.92 (1.63 - 9.42)* | 3.92 (10.86)* | 1.97 (0.79)* |

PT: Preferred Term; N, the number of adverse event reports; *Signal detected, see "Methods" for the criteria of detection.

no statistically significant difference in time-to-onset of suicide between male and female recipients (p = 0.5 > 0.05). Weibull distribution fitting was applied to the seven reports with available onset data. The shape parameter (β) for the onset time of suicide was estimated to be 0.70, with a 95% CI of 0.28–1.11, suggesting the risk was considered constant over time, corresponding to a random failure-type curve.

## Discussion

### Analysis of results

This pharmacovigilance study provides a comprehensive analysis of suicide-related adverse events (AEs) associated with zopiclone and eszopiclone, utilizing data from the extensive FAERS database and corroborated by the JADER and

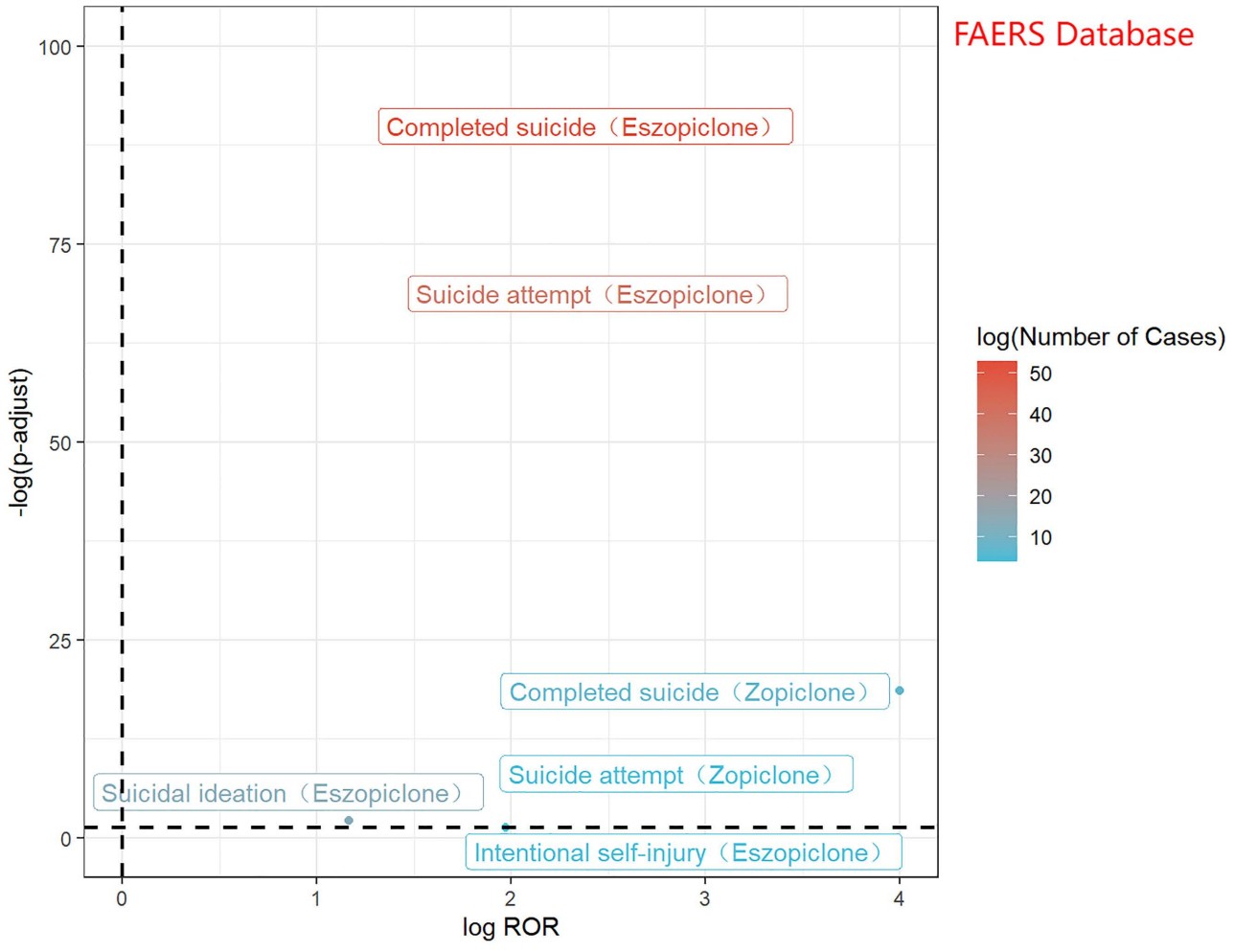

**Fig 3. Volcano map of zopiclone and eszopiclone-associated suicide logROR from FEARS.**

**Table 4. Clinical priority analysis of zopiclone and eszopiclone-associated suicide from FEARS.**

| Drug | PT | Clinical relevance | Reporting rate | Fatality rate (%) | Total score | relevant prioritization |
|------|-----|------|------|------|------|------|
| Zopiclone | suicide attempt | IME | 1.29% | 0 | 4 | Moderate priority |
| | completed suicide | IME | 2.29% | 100% | 6 | High priority |
| Eszopiclone | suicide attempt | IME | 1.05% | 0 | 4 | Moderate priority |
| | completed suicide | IME | 1.43% | 100% | 6 | High priority |
| | suicidal ideation | IME | 0.35% | 0 | 3 | Moderate priority |
| | intentional self-injury | IME | 0.13% | 0 | 3 | Moderate priority |

CVARD databases. The primary finding of this research is the detection of significant disproportionality signals for suicide attempts, completed suicide, and suicidal ideation linked to both zopiclone and eszopiclone. These findings were consistent across all three databases and were identified using multiple established signal detection methodologies (ROR, PRR, and IC), lending robustness to the observed associations.

**Table 5. The univariable logistic regression and multivariable logistic regression analysis of factors influencing the suicide associated with zopiclone and eszopiclone.**

| | Univariable | | | Multivariable | | |
|---|---|---|---|---|---|---|
| | OR | 95% CI | P | OR | 95% CI | P |
| **Age(years)** | | | | | | |
| <18 | 1.00 (Reference) | – | – | – | – | – |
| 18-65 | 716075.34 | 0.00-NA | p=0.994 | – | – | – |
| >65 | 295308.70 | 0.00-NA | p=0.994 | | | |
| **Gender** | | | | | | |
| Female | 1.00 (Reference) | – | – | – | – | – |
| Male | 14.70 | 2.64-274.75 | p=0.012* | 19.17 | 3.02-380.44 | p=0.009* |

The exposure factors considered included gender and age. OR indicates odds ratio."*" indicates significant, P<0.05.

**Table 6. Detection of suicide associated with zopiclone and eszopiclone from JADER/ CVARD.**

| | PT | N | ROR (95% CI) | PRR (2) | IC (IC$_{025}$) |
|---|---|---|---|---|---|
| **CVARD database** | | | | | |
| Zopiclone | suicidal ideation | 70 | 4.21 (3.32 - 5.33)* | 4.19 (167.89)* | 2.05 (0.39)* |
| | suicide attempt | 67 | 10.33 (8.09 - 13.18)* | 10.28 (543.15)* | 3.32 (1.65)* |
| | completed suicide | 45 | 9.19 (6.83 - 12.37)* | 9.16 (317.69)* | 3.16 (1.49)* |
| | suicidal behaviour | 5 | 10.43 (4.28 - 25.45)* | 10.43 (41.21)* | 3.34 (1.66)* |
| | self-injurious ideation | 6 | 5.47 (2.44 - 12.27)* | 5.47 (21.54)* | 2.43 (0.76)* |
| Eszopiclone | suicide attempt | 4 | 707.14 (224.98 - 2222.63)* | 518.84 (2064.4)* | 9.02 (7.2)* |
| **JADER database** | | | | | |
| Zopiclone | suicide attempt | 11 | 6.51 (3.59 - 11.81)* | 6.46 (50.55)* | 2.68 (1.02)* |
| | completed suicide | 11 | 14.23 (7.83 - 25.84)* | 14.1 (132.47)* | 3.8 (2.13)* |
| | suicidal ideation | 3 | 3.95 (1.27 - 12.29)* | 3.94 (6.57)* | 1.98 (0.31)* |
| Eszopiclone | suicide attempt | 10 | 12.52 (6.69 - 23.44)* | 12.31 (103.58)* | 3.62 (1.94)* |
| | suicidal ideation | 6 | 16.77 (7.48 - 37.59)* | 16.59 (87.44)* | 4.04 (2.37)* |
| | completed suicide | 3 | 8.04 (2.58 - 25.05)* | 8 (18.33)* | 3 (1.32)* |

PT: Preferred Term; N, the number of adverse event reports; *Signal detected, see"Methods"for the criteria of detection.

A key strength of this study is its multi-database approach, including a large primary analysis cohort from FAERS (initially 21,838,627 reports from Q1 2004 to Q4 2024, yielding 120 relevant cases) and external validation cohorts from JADER (44 reports) and CVARD (197 reports). This triangulation of evidence from different populations and reporting systems mitigates some of the biases inherent in single-database studies and strengthens the likelihood of a true association. The annual increase in the number of submitted reports, as illustrated in Fig 2, may reflect increased awareness, drug utilization, or a genuine rise in AEs over time, warranting continuous monitoring. The detailed flowchart (Fig 1) transparently outlines the data extraction and analysis process.

The descriptive analysis from FAERS revealed that nearly half of the 120 suicide-related reports involved female patients (46.7%), and the most affected age group was 18–64 years (53.3%). Alarmingly, death was reported as an outcome in 50% of these cases, underscoring the severity of these AEs. The United States and Japan were the predominant reporting countries. Furthermore, the clinical prioritization consistently classified "completed suicide" as a high-priority AE for both drugs in FAERS and generally moderate to high in JADER/CVARD, emphasizing the clinical relevance of these signals. This supports the critical role of post-marketing surveillance in identifying and characterizing rare but serious AEs.

**Table 7. Clinical priority analysis of suicide associated with zopiclone/eszopiclone from JADER/ CVARD.**

| Drug | PT | Clinical relevance | Reporting rate | Fatality rate (%) | Total score | relevant prioritization |
|---|---|---|---|---|---|---|
| **CVARD database** | | | | | | |
| Zopiclone | Suicidal ideation | IME | 0.55% | 11.43% | 3 | Moderate priority |
| | Suicide attempt | IME | 0.51% | 0 | 3 | Moderate priority |
| | Completed suicide | IME | 0.35% | 100% | 5 | Moderate priority |
| | Suicidal behaviour | IME | 0.02% | 33.33% | 4 | Moderate priority |
| | Self-injurious ideation | None | 0.05% | 16.67% | 2 | low priority |
| Eszopiclone | suicide attempt | IME | 28.57% | 0 | 5 | Moderate priority |
| **JADER database** | | | | | | |
| Zopiclone | Suicide attempt | IME | 0.87% | 0 | 3 | Moderate priority |
| | Completed suicide | IME | 0.96% | 100% | 5 | Moderate priority |
| | Suicidal ideation | IME | 0.26% | 33.33% | 4 | Moderate priority |
| Eszopiclone | Suicide attempt | IME | 1.74% | 0 | 4 | Moderate priority |
| | Suicidal ideation | IME | 1.04% | 0 | 4 | Moderate priority |
| | Completed suicide | IME | 0.52% | 100% | 5 | Moderate priority |

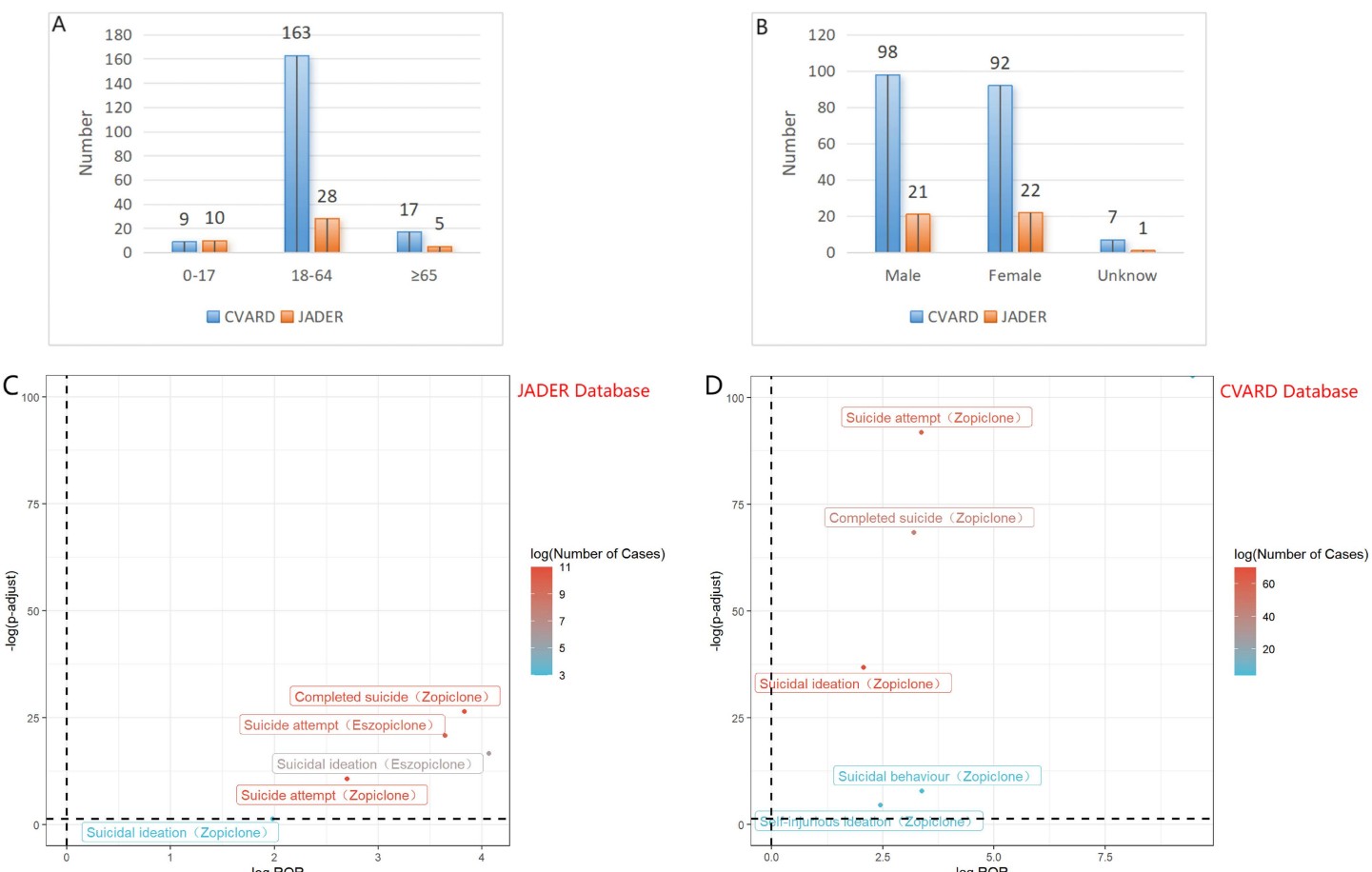

**Fig 4. The figure present statistical information on age and gender of reported patients(A and B), volcano map of suicide associated with zopiclone/eszopiclone logROR from JADER/ CVARD(C and D).**

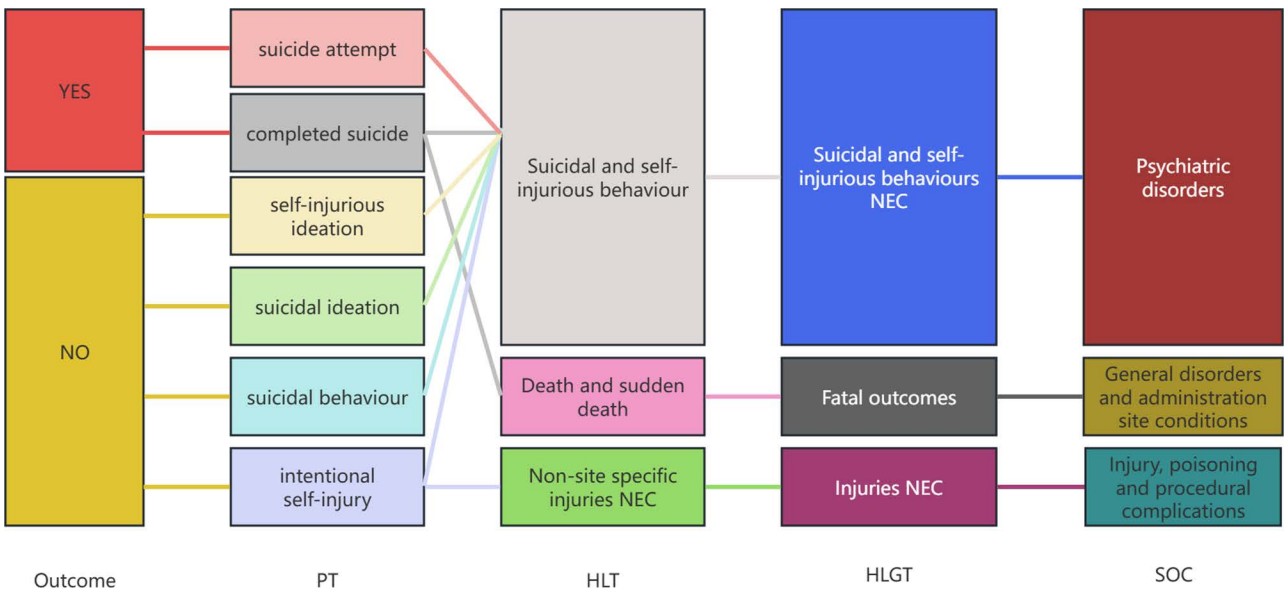

**Fig 5. The affiliation of suicide-related adverse events associated with zopiclone/eszopiclone in the MedDRA.**

The logistic regression analysis in FAERS provided crucial insights into risk factors associated with fatal outcomes among the reported suicide cases. While age was not found to be an influencing factor for suicide, gender emerged as significant. Specifically, multivariate analysis revealed that male patients had a strikingly higher risk of death compared to females (adjusted OR = 19.17, 95% CI = 3.02–380.44, P = 0.009) within this cohort of suicide-related AE reports. Although the wide confidence interval suggests some imprecision, likely due to the specific sample characteristics in the regression model, the finding is statistically significant and warrants further investigation into potential gender-specific vulnerabilities or reporting differences in fatal outcomes.

The time-to-onset (TTO) analysis, though based on a limited number of reports (N = 10 in FAERS, N = 7 in JADER/CVARD), suggested that the majority of suicide cases in FAERS (60%) occurred within 0–180 days post-administration. The median TTO was 155.5 days in FAERS and 19.8 days in JADER/CVARD. The Weibull distribution analysis indicated a shape parameter (β) of approximately 0.80 in FAERS and 0.70 in JADER/CVARD, suggesting a relatively constant risk over time (random failure-type curve). No statistically significant difference in TTO was observed between genders in either FAERS (p = 0.15) or JADER/CVARD (p = 0.5). The small sample sizes for TTO analyses necessitate caution in generalizing these specific temporal patterns. Similarly, the dosage analysis, with only 5 valid zopiclone cases and 30 eszopiclone cases from FAERS, provides preliminary data but is limited by sample size. The median zopiclone dose of 262.5 mg appears notably high and may reflect specific circumstances such as overdose, cumulative intake, or data entry variations, requiring careful interpretation.

The disproportionality signals were visualized using volcano plots (Fig 3 for FAERS, Fig 4C-4D for JADER/CVARD), which clearly depict adverse events with significant signal strength (higher on the -log10(P-value) axis) and magnitude of association (further along the log(ROR) axis). The MedDRA hierarchy (Fig 5) effectively illustrates the relationship between the various reported preferred terms and broader systemic effects.

Despite the robust signals, several limitations inherent to pharmacovigilance studies must be acknowledged. These include potential reporting biases (such as under-reporting or selective reporting of more severe outcomes), the inability to definitively establish causality (as associations may be confounded by indication, such as underlying depression or

anxiety for which these hypnotics were prescribed), the CVARD is unable to statistically analyze TTO data (hence the external severity only analyzed the TTO data of JADER).and variations in the quality and completeness of AE reports. The information regarding missing demographic data in Table 3 further highlights this.

### Studies suicide associated with zopiclone/eszopiclone

Zopiclone, eszopiclone, and benzodiazepines exhibit common pharmacological effects by acting on GABAA receptors. Clinical studies have demonstrated that benzodiazepines can effectively induce sedation in patients, but some patients experience diametrically opposite results, a phenomenon known as a paradoxical reaction. Patients may exhibit suicidal, aggressive, or larcenous behaviours even if they do not have a history of similar behaviour prior to drug administration [24].

The ability of nonbenzodiazepines to alleviate anxiety and produce similar depressive effects could explain patients'suicidal tendencies through (1) blocking receptors in the brain from taking up and clearing serotonin, thereby increasing serotonin levels and alleviating anxiety [10], and (2) elevating γ-aminobutyric acidergic transmitters in the brain, which not only relieves anxiety with prolonged drug use but also contributes to pharmacological depression.

### Limitation

Limitations of this study include risk bias due to the following factors: (1) patients with concomitant depression who are prescribed eszopiclone or zopiclone also have an increased risk of suicide [25]; (2) the FAERS and JADER database have only been publicly available since 2004; therefore, reports of suicide-related adverse events caused by zopiclone/eszopiclone use in patients prior to 2004 are missing; (3) We retrieved 32 reports from Japan and 3 reports from Canada in the FAERS database, which may overlap with reports in the JADER and CVARD databases, respectively; and (34) the European Medicines Agency approved the marketing of eszopiclone in Europe in March 2020, with only French pharmaceutical production being authorized. Eszopiclone is not yet widely used in other European countries, resulting in fewer adverse events reported from European countries in the FAERS database. Furthermore, zopiclone has not been approved by the FDA for marketing in the U.S., leading to a lack of zopiclone-related adverse event reports from this country in the FAERS database.

### Conclusion

Pharmacovigilance analyses of the FAERS database showed that zopiclone and eszopiclone are linked to heightened susceptibility to suicide attempt and completed suicide. Therefore, it is recommended that patients, particularly those at potential risk of suicide, be evaluated for possible risks and benefits. Furthermore, additional studies are necessary to elucidate the potential mechanisms by which nonbenzodiazepines cause suicide-related adverse events. Given that these events are serious adverse effects, randomized controlled trials with large sample sizes are needed to guide the future use of these drugs.

### Supporting information

**S1 Table. Two-by-two contingency table for disproportionality analyses.**
(DOCX)

**S2 Table. Four major algorithms used for signal detection.**
(DOCX)

**S3 File. The data involved in the research.**
(ZIP)

## Acknowledgments

Thank you to all the medical staff who contributed to the maintenance of the medical record database (FAERS,JADER and CVARD).

## Author contributions

**Conceptualization:** Fang Li, Mei Zhang.

**Data curation:** Fang Li, Mei Zhang, Li Huang.

**Investigation:** Yinpeng Xu, Fang Li, Li Huang.

**Methodology:** Fang Li, Li Huang.

**Project administration:** Li Huang.

**Supervision:** Li Huang.

**Validation:** Li Huang.

**Writing – original draft:** Yinpeng Xu.

**Writing – review & editing:** Yinpeng Xu, Fang Li, Mei Zhang, Li Huang.

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
