## [Decision Letter · Decision Letter 0]

24 Nov 2025

Dear Dr. Huang, 

We look forward to receiving your revised manuscript.

Kind regards,

James M Wright

Academic Editor

PLOS ONE

Journal Requirements:

3. Please note that funding information should not appear in the Acknowledgments section or other areas of your manuscript. We will only publish funding information present in the Funding Statement section of the online submission form. Please remove any funding-related text from the manuscript. 

4. Please ensure that you refer to Figure 4 in your text as, if accepted, production will need this reference to link the reader to the figure.

6. We note that there is identifying data in the Supporting Information file <The data involved in the research.zip>. Due to the inclusion of these potentially identifying data, we have removed this file from your file inventory. Prior to sharing human research participant data, authors should consult with an ethics committee to ensure data are shared in accordance with participant consent and all applicable local laws.

-Location data

Please remove or anonymize all personal information, ensure that the data shared are in accordance with participant consent, and re-upload a fully anonymized data set. Please note that spreadsheet columns with personal information must be removed and not hidden as all hidden columns will appear in the published file.

7. We are unable to open some files (.R, RDATA, RHISTORY file format) in your Supporting Information file [The data involved in the research.zip]. Please kindly revise as necessary and re-upload.

8. Thank you for providing your underlying data as Supporting Information.

We note that the data set contains text or data that is not in English. Please note that PLOS is an English-language publisher, so we require data sets to be provided in English as well. Please upload an English-language version of your data set.

This will also allow us to determine if your data follows PLOS standards per our Data Availability policy here: https://journals.plos.org/plosone/s/data-availability 

**Additional Editor Comments:**

This represents an important analysis of suicide related events associated with zopiclone and eszopiclone.

1. the term post-vaccination is used inappropriately. Please correct throughout

2. Some Tables and Figures are based on very small numbers and are therefore misleading. Please remove Fig.4, 5 and 8 and Table 7 and 11. The written description of this data is sufficient.

3. Table 8 should be removed as it does not provide any new information already provided in the text.

Reviewers' comments:

Reviewer's Responses to Questions

**Comments to the Author**

1. Is the manuscript technically sound, and do the data support the conclusions?

Reviewer #1: Yes

2. Has the statistical analysis been performed appropriately and rigorously?

Reviewer #1: Yes

3. Have the authors made all data underlying the findings in their manuscript fully available?

Reviewer #1: Yes

4. Is the manuscript presented in an intelligible fashion and written in standard English?

Reviewer #1: Yes

Reviewer #1: This pharmacovigilance analysis on Zopiclone and Eszopiclone is well constructed and presented. The objective is articulated and supported by a well-defined rationale that addresses a relevant gap in the field.

My comment on the title would be since you mainly analyzed data from FAERS and used data from JADER and CVARD to corroborate the findings, consider omitting the latter from the title, as it might give a different impression.

If you could justify the age categories (as 18 to 65 is a very wide group)?

Were there other exclusion or inclusion criteria used?

Otherwise methodological framework including signal detection and statistical validation is rigorous.

Results are clearly presented, logically structured, and supported by appropriate statistical interpretation.

The discussion shows balanced overview of the findings, including acknowledgment of study limitations.

Overall, this is a well-constructed, methodologically sound, and scientifically valuable contribution to the field.

**Do you want your identity to be public for this peer review?** For information about this choice, including consent withdrawal, please see our Privacy Policy

Reviewer #1: No

---

## [Author Response · Author response to Decision Letter 1]

16 Dec 2025

Response to Reviewers and Editor

1.My comment on the title would be since you mainly analyzed data from FAERS and used data from JADER and CVARD to corroborate the findings, consider omitting the latter from the title, as it might give a different impression.

Reply: Dear Reviewer, although FAERS served as the primary data source, the incorporation of JADER and CVARD was not merely supplementary, but rather provided meaningful corroboration across different pharmacovigilance systems, enhancing the robustness and generalizability of our findings.

Therefore, including all three databases in the title accurately reflects the multi-database design of the study and supports the intended scope and scientific contribution. To maintain clarity and consistency with the study’s objectives and methodology, we prefer to retain the current title.

2.If you could justify the age categories (as 18 to 65 is a very wide group)?

Were there other exclusion or inclusion criteria used?

Reply: Thank you for this comment. We categorized age into these three groups (<18, 18–64, and ≥65) to align with standard pharmacovigilance practices that distinguish between pediatric, general adult, and geriatric populations. In our analysis, the "18–64" group represents the general adult population. We considered subdividing this group; however, given the total sample size of our study (e.g., n=64 in the 18–64 group for FAERS), further stratification would have resulted in subgroups too small to yield statistically robust results in the logistic regression analysis. Therefore, we maintained the broader categories to ensure statistical power.

regarding the criteria, this study was a retrospective analysis of spontaneous reporting databases (FAERS, JADER, CVARD). Our inclusion criteria were strictly defined by the presence of the suspect drugs (Zopiclone/Eszopiclone) and specific MedDRA Preferred Terms (PTs) related to suicide. We did not apply additional clinical exclusion criteria (such as excluding specific comorbidities) because spontaneous reporting systems often lack complete clinical history, and we aimed to reflect real-world reporting profiles. However, we did apply rigorous data cleaning, including the removal of duplicates and reports with missing critical data, which served as our exclusion controls.

3.Please ensure that your manuscript meets PLOS ONE's style requirements, including those for file naming.

Reply: We have carefully reviewed and revised our manuscript, including the main body and the title/authors/affiliations section, to ensure full compliance with the current PLOS ONE style templates and file naming conventions as outlined in the provided guidelines.The revised files have been uploaded for your review.

4.We note that the grant information you provided in the ‘Funding Information’ and ‘Financial Disclosure’ sections do not match. When you resubmit, please ensure that you provide the correct grant numbers for the awards you received for your study in the ‘Funding Information’ section.

Reply: We apologize for the oversight regarding the grant information.We have thoroughly reviewed the 'Funding Information' and 'Financial Disclosure' sections and have corrected the grant numbers to ensure that the details are consistent and accurate in both places.The revised manuscript now contains the correct and matching funding information as required.

5.Please note that funding information should not appear in the Acknowledgments section or other areas of your manuscript. We will only publish funding information present in the Funding Statement section of the online submission form. Please remove any funding-related text from the manuscript.

Reply: We have carefully reviewed the manuscript and confirm that all funding-related text has been removed from the Acknowledgments section and any other part of the manuscript file.

We have ensured that the complete and accurate funding information is present exclusively in the designated 'Funding Statement' section of the online submission form, as per your journal's policy. The revised manuscript has been uploaded.

6.Please ensure that you refer to Figure 4 in your text as, if accepted, production will need this reference to link the reader to the figure.

Reply:We have carefully reviewed the manuscript and confirmed that Figure 4 has been clearly cited in the text as required. However, in the editor's supplementary comments, considering that the sample size of Figure 4 is small and may be misleading, it was suggested that Figure 4 be deleted. Therefore, this image has been removed.

7.Please include captions for your Supporting Information files at the end of your manuscript, and update any in-text citations to match accordingly.

Reply:We have thoroughly reviewed the manuscript and have included captions for S1 and S2 Table Supporting Information files at the end of the main text. We have also updated all in-text citations to ensure they match the newly formatted Supporting Information sections.

8.We note that there is identifying data in the Supporting Information file <The data involved in the research.zip>. Due to the inclusion of these potentially identifying data, we have removed this file from your file inventory. Prior to sharing human research participant data, authors should consult with an ethics committee to ensure data are shared in accordance with participant consent and all applicable local laws.

Data sharing should never compromise participant privacy. It is therefore not appropriate to publicly share personally identifiable data on human research participants. Please remove or anonymize all personal information, ensure that the data shared are in accordance with participant consent, and re-upload a fully anonymized data set.

Reply: We fully understand and respect the strict requirements regarding participant privacy and data ethics.We would like to clarify that the data analyzed in this study were retrieved from public pharmacovigilance databases (FAERS, JADER, and CVARD) , which are publicly accessible resources intended for post-marketing safety surveillance. While these datasets are generally de-identified by the respective regulatory authorities, we acknowledge your concern that certain data points in our raw file might still be considered potentially identifying.

In accordance with your instructions, we have thoroughly reviewed the dataset and removed all information that could potentially lead to identification (including specific case identifiers and unmasked demographic details). We have re-uploaded the fully anonymized dataset. We confirm that the shared data strictly adheres to ethical standards and protects participant privacy.

9.We are unable to open some files (.R, RDATA, RHISTORY file format) in your Supporting Information file [The data involved in the research.zip]. Please kindly revise as necessary and re-upload.

Reply: Thank you for your feedback regarding the Supporting Information files. The files in .R, .RDATA, and .RHISTORY formats are standard formats generated by R statistical software. To open these files, users would need to install R, RStudio, and Rtools, which are freely available and widely used for data processing and statistical analysis.

The .R file contains the complete script (the source code) used for data cleaning and performing the statistical analyses (ROR, PRR, IC).

The .RDATA file is the saved workspace containing the processed data objects.

The .RHISTORY file is an auxiliary log file documenting the commands run.

We would also like to clarify that the .R file contains the primary analysis code, while the extracted datasets generated from these scripts have already been provided in the supplementary materials in fully accessible formats (Excel/CSV). Therefore, access to the code files is not essential unless further replication or validation of the analysis is specifically required.

Given that all necessary output data are already included in accessible formats, it may not be necessary to open the R-based files unless detailed code-level review is desired.However, we are happy to keep or remove these files according to the journal’s preference.

10.We note that the data set contains text or data that is not in English. Please note that PLOS is an English-language publisher, so we require data sets to be provided in English as well. Please upload an English-language version of your data set.

Reply: We apologize for the oversight. We have thoroughly reviewed and translated all non-English text and data fields within the Supporting Information file into English, including the necessary headers and text-based entries derived from the JADER/CVARD systems.We have re-uploaded the fully English-language version of the data set, ensuring full compliance with PLOS publishing requirements.

11.the term post-vaccination is used inappropriately. Please correct throughout

Reply: Thank you for pointing out the inappropriate use of the term "post-vaccination" in our manuscript. This was a clear oversight on our part.We have carefully reviewed the entire manuscript and replaced every instance of "post-vaccination" with the correct and precise terminology “after taking medication”.

12.Some Tables and Figures are based on very small numbers and are therefore misleading. Please remove Fig.4, 5 and 8 and Table 7 and 11. The written description of this data is sufficient.

Reply: We agree that the descriptive text sufficiently covers the findings based on small numbers. In accordance with your guidance, we have removed the following figures and tables from the manuscript:

Figures: Fig. 4, Fig. 5, and Fig. 8

Tables: Table 7 and Table 11

We have carefully updated the main text and numbering of the remaining figures and tables to ensure consistency throughout the manuscript.Thank you again for helping us improve the clarity and focus of our presentation.

13.Table 8 should be removed as it does not provide any new information already provided in the text.

Reply:In accordance with your instruction, we have removed Table 8 from the manuscript, as the information was already adequately presented in the main text.

---

## [Editor Report · Decision Letter 1]

21 Dec 2025

Suicide adverse events associated with zopiclone and eszopiclone: A pharmacovigilance analysis based on FAERS, JADER and CVARD

PONE-D-25-37985R1

Dear Dr. Li Huang,

We’re pleased to inform you that your manuscript has been judged scientifically suitable for publication and will be formally accepted for publication once it meets all outstanding technical requirements.

Kind regards,

James M Wright

Academic Editor

PLOS One
---

## [Editor Report · Acceptance letter]

PONE-D-25-37985R1

PLOS One

Dear Dr. Huang,

I'm pleased to inform you that your manuscript has been deemed suitable for publication in PLOS One. Congratulations! Your manuscript is now being handed over to our production team.

Kind regards,

on behalf of

Professor James M Wright

Academic Editor

PLOS One